# CuPID: Leveraging Masked Single-Lead ECG Modelling for Enhancing the Representations

## Abstract

Wearable sensing devices, such as electrocardiogram (ECG) heart-rate monitors, will play a crucial role in the future of digital health. This continuous monitoring leads to massive unlabeled datasets, making the development of unsupervised learning frameworks essential to associate these single-lead ECG signals to their anticipated clinical outcomes. While the Masked Data Modelling (MDM) methods have enjoyed wide use, the idiosyncrasies of single-lead ECG data make its direct application impractical. In this paper, we present Cueing the Predictor Increments the Detailing (CuPID), a novel Self-Supervised Learning (SSL) method that adapts MDM methods for use on single-lead ECG data. CuPID accomplishes this via cueing spectrogram-derived context to the predictors, thus incentivizing the encoder to produce more detailed representations. This leads the class token to accommodate fine-grained information. We demonstrate that CuPID outperforms state-of-the-art methods in a variety of downstream tasks and databases, increasing the accuracy for each task from 3.6 % to 9.7%.

## 1 Introduction

The wearable sensing field has seen remarkable advancements in recent years, and is expected to play a crucial role in the future of digital health. One widely used type of wearable health sensor is the heart monitor that captures cardiac activity as single-lead ECG signals during free-living conditions, such as in the patient's home. Mapping these signals with significant clinical outcomes has the potential to provide outstanding benefits such as simplifying the diagnostic process (Himmelreich et al., 2019) or enabling users to engage proactively in tracking their heart health (Abdou & Krishnan, 2022). In this context, models that extract information from single-lead ECG into generalizable representations are mandated to address distinct downstream tasks. These models should be optimized using large volumes of unlabelled data. This makes Self-Supervised Learning (SSL) framework particularly well-suited for addressing this clinical challenge.

Recently, Masked Data Modelling (MDM) methods have been gaining attention in the SSL field (He et al., 2021; Gupta et al., 2023; Assran et al., 2023). They rely on masking a portion of the input and driving a transformer-based encoder, typically a Vision Transformer (ViT) (Dosovitskiy et al., 2021) to compute detailed patch representations that enable a predictor to infer the information accommodated within the unseen patches. This approach is especially effective in fields like computer vision, where the predictor can associate unseen tokens with the object they represent by simply perceiving a portion of the whole picture and having the spatial information of unseen patches.

However, it is impractical to directly apply these methods to single-lead ECG data. These signals capture the sequence of activities that are executed in each beat by the heart's different chambers to ensure the blood reaches the entire body. Figure 1a illustrates how various cardiac activities are represented by distinct wave morphologies in the ECG. Even though the sequence of activities occurs periodically over time, the distance between consecutive periods varies moderately as shown in Figure 1b and 1c, respectively. This combination of ECG idiosyncrasies leads to the following dilemma: it is challenging for the predictor to accurately model the position of each wave for masked inputs because of the varying distances between periods, and not inferring exactly this position has a big impact on the loss since consecutive strips accommodate distinct waves. This dilemma leads the predictor to be cautious when reconstructing the masked patches. As shown in Figure 2b, it prefers to estimate a value near the average rather than trying to match precisely the signal's morphology.

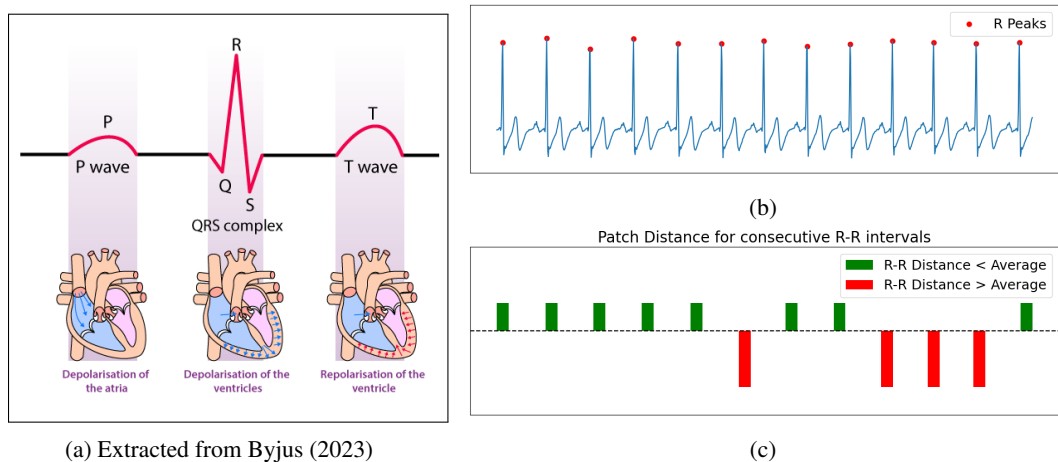

(a) Extracted from Byjus (2023)                    (c)

Figure 1: The different heart actions and their corresponding morphology in the ECG are detailed in (a). The distance between R-R peaks in patches for a normal ECG (b) is displayed in (c)

This paper presents Cueing the Predictor Increments the Detailing (CuPID), which is a novel SSL method that addresses the previously mentioned issue by cueing the predictor with contextual information provided by the spectrogram of the input signal. This information is fed into the attention mechanism of the transformer-based predictor as the Key (K) to ensure that its role is merely informative and its value can not be used directly to reconstruct the representations. It leads to the loss function reaching significantly lower values, as shown in Figure 2a. Therefore, the reconstructions are more adjusted to the morphology of the original signal, as captured in Figure 2c. Although the CuPID predictor is provided with additional information, making these results insignificant on their own, we hypothesize that: (i)The predictor's inability to reconstruct the original signal due to the unpredictability of the distance between periods limits the encoder's learning potential. (ii) By cueing the predictor with the spectrogram, we enable it to manage this delay and drive the encoder to compute detailed token representations, which can be used to reconstruct the original input with high precision. (iii) The more informative the patch representations are, the more informative the class token will be, thereby enhancing the model's performance in downstream tasks.

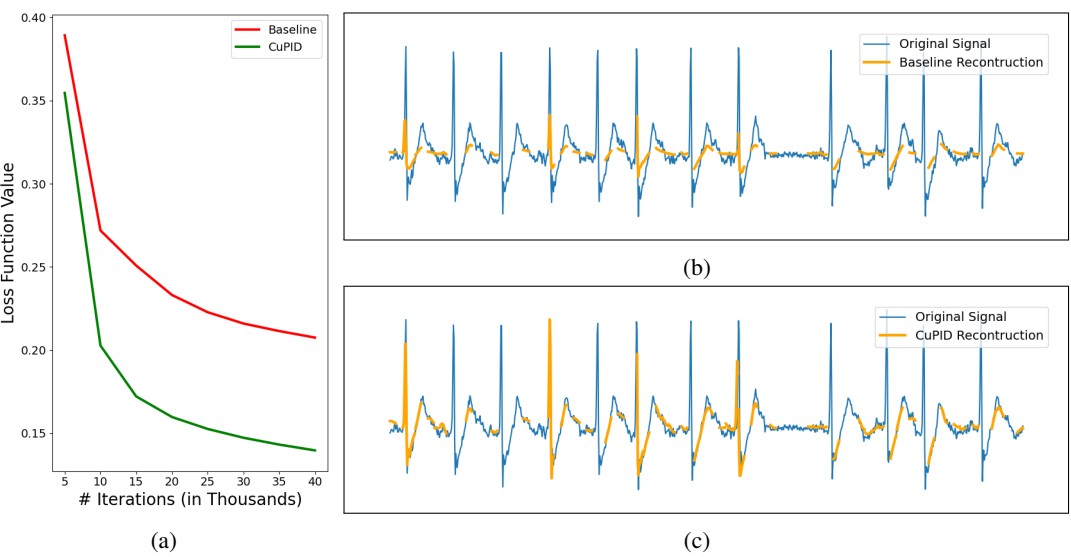

(a)                                  (c)

Figure 2: (a) Represents the evolution of the loss across the training procedure. (b) and (c) show that more accurate reconstructions are computed by the predictor when the spectrogram is incorporated.

To assess our hypothesis, we have conducted an extensive evaluation where CuPID is compared against the existing state-of-the-art (SOTA) SSL methods tailored for single-lead ECG analysis. In the proposed evaluation, up to three distinct noisy databases; MIT-BIH Atrial Fibrillation (MIT-AFIB) (Moody & Mark, 1983), MIT-BIH Supraventricular Arrhythmia (MIT-SVA) Greenwald et al. (1990), and Long Term AF (LT-AF) (Petrutiu et al., 2007), are considered. Additionally, CuPID has been evaluated on widely-used benchmarks, i.e., *PTB-XL* (Wagner et al., 2020), and *CPSC2018* (Alday et al., 2021) against the SOTA MDM methods tailored for 12-lead ECG processing. Remarkably, CuPID achieves significantly superior performance when compared with single-lead ECG methods. Additionally, it shows competitive performance compared to 12-lead ECG models, despite CuPID using a significantly smaller model and only one lead sampled at a lower resolution for inference. Finally, the benefit of incorporating the spectrogram has been assessed for different pre-training databases and different configurations.

In summary, the contributions of this paper are:

- We have discussed the limitations of applying MDM techniques directly to single-lead ECG signals due to the idiosyncrasy of this kind of data.
- We introduce CuPID, a novel SSL method that addresses these limitations by helping the predictor during the pre-training. This is made by incorporating the spectrogram of the input signal to the attention mechanism as the Key, limiting its role to be merely informative.
- We provide a model that achieves markedly enhanced results in a variety of downstream tasks that are relevant for cardiovascular remote monitoring.

## 2 RELATED WORK

### 2.1 MASKED DATA MODELLING (MDM)

Masked Data Modelling (MDM) has been a commonly used technique in the Natural Language Processing (NLP) field. Methods such as Bidirectional Encoder Representations from Transformers (BERT) (Devlin et al., 2019) that rely on hiding a series of words within a sentence and optimizing a predictor to infer these words have proven to be the most effective pre-training method in the field. In recent times, this pre-training mechanism has been adapted in the field of computer vision. Existing methods, such as, Masked Autoencoders (MAE) (He et al., 2021) or Siamese Masked Autoencoders (SiamMAE) (Gupta et al., 2023) incorporate a predictor trained to reconstruct masked patches from the original input. Alternatively, Image-based Joint-Embedding Predictive Architecture (I-JEPA) (Assran et al., 2023) reconstructs the representations computed by a teacher network instead of the input itself. The weights of this teacher network are not optimized using the gradients but by an exponential moving average (EMA) of the weights of the student network. Both approaches have shown promising results in the field of computer vision, outperforming gold-standard Energy-Based Modelling (EBM) methods such as Variance-Invariance-Covariance Regularization (VIC-REG) (Bardes et al., 2022), Self-Distillation with no Labels (DINO) (Caron et al., 2021), or Bootstrap Your Own Latent (BYOL) (Grill et al., 2020).

Given the idiosyncrasies of ECG data, we consider a more suitable to reconstruct the original input rather than the teacher representations. This is due to the fact that the critical information in ECG data resides in the morphology of each heartbeat. Reconstructing the original input ensures that these waves are given greater importance since the amplitude values are greater than strips with no waves. This interesting property does not occur when reconstructing the representations from the teacher network, since they are expected to lie with the same range of values. This is reflected in better-performing models by optimizing them to reconstruct the original input (See Section A). However, the effect of incorporating the spectrogram into the predictor has also been studied for the two approaches (See Section 5).

### 2.2 SSL IN SINGLE-LEAD ECG SIGNAL PROCESSING

Most-widely used single-lead ECG SSL methods follows a EBM approach; (i) Contrastive Learning of Cardiac Signals Across Space (CLOCS) (Kiyasseh et al., 2021) utilizes two consecutive ECG time strips as positive pairs, (ii) Mixing-Up (Wickstrøm et al., 2022) introduces a more tailored

data augmentation product of two time series from the same recording, (iii) Patient Contrastive Learning (PCLR) (Diamant et al., 2022) which considers two time strips from the same subject but different recordings. While all these methods utilize the Contrastive Learning (Chen et al., 2020) as a common framework for learning the invariant attributes considering non-overlapping inputs as positive pairs, (iv) Distilled Embedding for Almost-Periodic Time Series (DEAPS) (Atienza et al., 2024) drives the model to capture the also dynamic patterns of the single-lead ECGs. It follows a non-contrastive learning approach, being built on top of BYOL (Grill et al., 2020).

All of these SSL methods will compose the set of baselines for the CuPID's evaluation, where the representations computed by each pre-trained model will be employed for addressing several downstream tasks.

## 2.3 SSL in 12-Lead ECG Signal Processing

In the realm of 12-lead ECG signals, research has effectively utilized MDM techniques. The introduction of a new spatial dimension broadens the scope for input masking, thereby aiding the predictor in identifying the locations of various waves within the masked tokens. Techniques like MTAE, MLAE, and MLTAE, all introduced by MAE family of ECG (MaeFE) (Zhang et al., 2023), suggest three masking strategies: temporal masking, spatial masking across different leads, or a combination of both. More recent approaches, such as Spatio-Temporal Masked Electrocardiogram Modeling (ST-MEM) (Na et al., 2024), adopt this combined strategy by employing a joint predictor that reconstructs the original input attending to each lead independently.

Among the four listed methods, only MTAE is suitable for single-lead ECG signals, as the other three methods require multiple leads. Consequently, only MTAE has been trained to handle single-lead signals and has been included in the main evaluation. Nevertheless, CuPID is also benchmarked against these methods when 12-lead data is available, despite the fact that CuPID only utilizes one lead for inference.

## 3 Cueing the Predictor Increments the Detailing (CuPID)

The core idea behind CuPID is cueing the predictor with the contextual information provided by the spectrogram. Its workflow is illustrated in Figure 3. From left to right the original signal input is patched and embedded using a linear layer. A portion of these tokens (Represented as gray blocks in the figure) is randomly masked with a fixed ratio. Only the unmasked tokens are passed through the encoder. Learnable mask tokens with their respective positional encoding are placed in the original position of the masked segments. What sets CuPID apart is that it uses the spectrogram as the Key for the attention mechanism, as represented in Figure 3. This predictor reconstructs the original input. The $\mathcal{L}_1$ metric is computed between this reconstruction and the original input. This loss function is only calculated on the masked patches. It is important to note that the predictor is discarded after training, with the encoder being used for downstream tasks. Therefore, the spectrogram is only utilized during pre-training and not during inference.

### 3.1 Role of Spectrogram in the Predictor

The core idea behind CuPID is providing the predictor with more contextual information than the regular positional encodings. We identify the spectrogram as a tool that has the potential to do it. A spectrogram is a visual representation of the spectrum of frequencies in a signal as they vary over time. They are commonly generated using the Fast Fourier Transform (FFT), which converts a time-domain signal into its frequency components. The spectrogram is expected to provide detailed contextual information to the predictor since the waves composing the R-R interval operate in distinct frequencies. This feature is leveraged by traditional signal processing methods to perform ECG signal delineation (Martinez et al., 2004).

**Limiting the Information Provided by the Spectrogram:** Just as the time domain input is transformed into the frequency domain when computing the spectrogram, it can also be converted back to the time domain. It means that the predictor could potentially reconstruct the original input without using the encoder's representations. To prevent this, the spectrogram is used just as the K in the

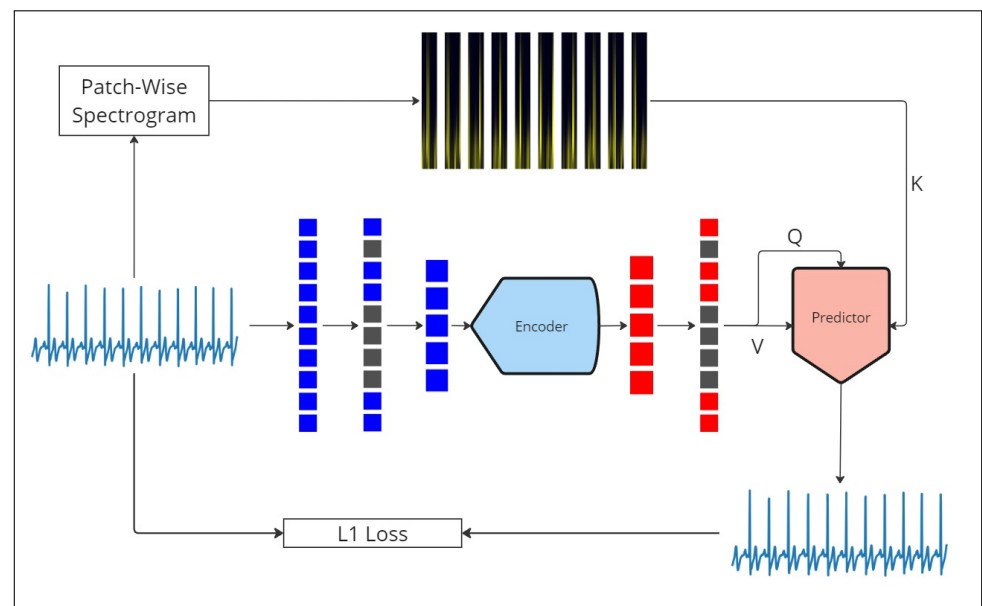

Figure 3: CuPID architecture. The proposed method mirrors the standard framework for MDM approaches. The incorporation of the spectrogram into the predictor's attention mechanism sets CuPID apart. The encoder is the model used to address the downstream tasks, while the predictor is discarded after the pre-training. Therefore, this spectrogram is not provided during the evaluation.

attention mechanism when fed into the predictor. This transformer-based predictor relies on the standard attention mechanism formulated on Vaswani et al. (2017). It is composed of three components, i.e., query ($Q$), key ($K$), and value ($V$) and it is expressed as the following:

$$\text{Attention}(Q, K, V) = \text{softmax}\left(\frac{QK^T}{\sqrt{d_k}}\right)V, \tag{1}$$

where the query ($Q$) refers to the token that is attending the others for information, the key ($K$) represents what information can be found in the specific token, and the value ($V$) accommodates the information. It is worth highlighting that the $K$ only has the potential of informing what kind of information it could be found in the respective token, but not providing information by itself. This information is provided by the corresponding $V$. In other words, even though the spectrogram gathers all the information needed for reconstructing the input, this information can not be applied directly.

**Challenges of Using the Spectrogram as the Key:** A primary issue arises when using the spectrogram as the key instead of the standard concatenation of encoder representations and mask tokens. The predictor cannot distinguish between informative tokens and mask tokens, as this distinction is not present in the spectrogram. It is important to note that CuPID, in accordance with standard practices, permits mask tokens to interact with each other, ensuring the spectrogram remains unmasked. Consequently, a token might incorrectly focus on another due to its spectrogram key, even if that token is merely a mask and contains no actual information. To overcome this issue, CuPID delays incorporating the spectrogram into the predictor's second block. The regular concatenation of encoder representations and mask tokens is used as the K in the first block. This approach ensures that each mask token retains some information after the initial block, which can then be distilled in subsequent blocks with the context information provided by the spectrogram as K.

Considering these two crucial aspects, Figure 4 depicts the CuPID predictor. In the initial block, the inference follows a conventional approach, while the spectrogram is integrated into subsequent blocks as the $K$ in the attention mechanism. This predictor computes the single-lead ECG reconstruction, which is compared to its corresponding original input using the $\mathcal{L}_1$ metric. This metric serves as the sole loss function of the model and is represented by the following formula:

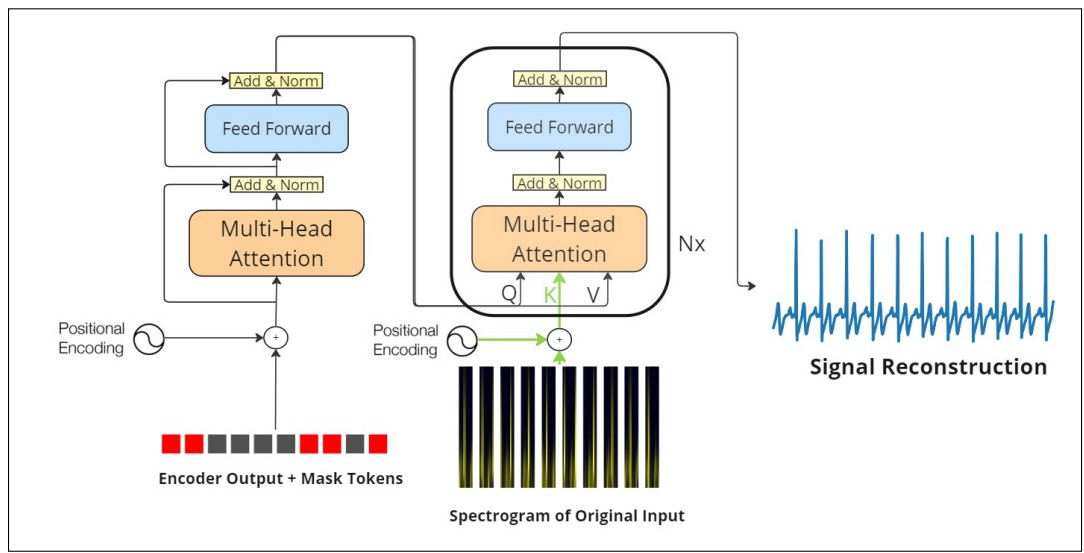

Figure 4: Diagram of CuPID's Predictor. Due to the challenges of using the spectrogram as a Key, the spectrogram is incorporated from the second block of the predictor. Its first block mirrors the standard predictor block for MDM framework.

$$\mathcal{L}_1(X, \hat{Y}, \mathcal{M}) = \frac{1}{sum(\mathcal{M})} \cdot \sum_{i=1}^{n} \left| Y_i - \hat{Y}_i \right| \cdot \mathcal{M}_i, \tag{2}$$

where $X, \hat{Y}, \mathcal{M}$, and $n$ represent the original input, the predictor reconstruction, the mask, and the number of patches, respectively.

## 3.2 IMPLEMENTATION DETAILS

To ensure the replication of the method, we meticulously outline the hyperparameter settings and the model architecture.

**Model Architecture:** The ViT model proposed by CuPID for processing the single-lead ECG signals counts with four regular transformer blocks with four heads each and a dimension of 128. The input consists of a one-dimensional 10-second signal sampled at 100 Hz. This signal is split into patches with a length of 10 samples.

**CuPID Implementation and Optimization**: The predictor consists of a ViT model with two blocks and a dimension of 128. The training procedure consists of 40,000 iterations. We use a batch size of 256, AdamW (Loshchilov & Hutter, 2019) optimizer with a learning rate of $1e-3$. The masking ratio is set to 0.5. To achieve the patch-wise spectrogram consistent with the dimensions of the predictor, the number of coefficient bins is set to 255, and the window length to 20. CuPID has been trained on publicly available Sleep Heart Health Study (SHHS) database (Zhang et al., 2018; Quan et al., 1998). The training procedure and the evaluations are performed on a desktop computer, with a Nvidia GeForce RTX 3070 GPU.

The influence of the masking ratio parameter as well as the influence of incorporation of the spectrogram for different values of it has been evaluated (See Section 5). In addition, the effect of incorporating the spectrogram has also been studied in the Icentia (Tan et al., 2019) database.

# 4 EVALUATION

## 4.1 COMPARISON AGAINST SOTA

To assess the ability of the method to generalize different classes within the same record, given a limited number of labeled noisy recordings from Holter monitors, CuPID has been evaluated against the following methods that compel the set of baselines for the evaluation; CLOCS (Kiyasseh et al., 2021), PCLR (Diamant et al., 2022), Masked Time Autoencoder (MTAE), from Zhang et al. (2023); DEAPS (Atienza et al., 2024); and Mix-up (Wickstrøm et al., 2022). In addition, a version of Image-based Joint-Embedding Predictive Architecture (I-JEPA) tailored for processing 1-D ECG input has been included. To ensure fairness in the evaluation, all the methods have been trained using the same training configuration, encoder, and dataset as CuPID. The objective is to develop a generic model whose representations can be directly used on multiple downstream tasks. Therefore, our experiments are focused on linear evaluation. We have carried out the following experiments on the following databases; MIT-AFIB (Moody & Mark, 1983); LT-AF (Petrutiu et al., 2007) and MIT-SVA (Greenwald et al., 1990). More details for each particular dataset are provided in the Appendix (See Section B). All these databases are publicly available on Physionet (Goldberger et al., 2000). The specifics of each experiment are detailed as the following:

**Atrial Fibrillation (AFib) Identification on MIT-BIH Atrial Fibrillation (MIT-AFIB):** This dataset accommodates long-term recordings of 23 subjects transitioning between Normal Sinus Rhythm (NSR) to paroxysmal AFib episodes and vice versa. We have conducted a Leave-One-Out (LOO) cross-validation across the 23 MIT-AFIB subjects, where a Support Vector Classificatier (SVC) (Platt, 2000) is fitted on top of the representations. We want to highlight that CuPID outperforms significantly all the baselines, as reflected in Table 1.

**Cardiovascular Arrhythmias Detection on Long Term AF (LT-AF):** This dataset compels long-term recordings of 84 subjects. It is composed of subjects suffering spontaneous bradycardia episodes and subjects with sustained AFib in addition to subjects suffering paroxysmal AFib episodes that are also contained in the previous dataset. We have repeated 10 times a 10-fold cross-validation across the 84 LT-AF subjects, where a Logistic-Regression model is fitted on top of the representations. Table 1 reflects that CuPID remarkably outperforms all the baselines.

**Abnormal Beat Identification on MIT-BIH Supraventricular Arrhythmia (MIT-SVA):** This database contains beat-wise annotators for Normal, Ventricular or Supraventricular beats. Since all methods used for this evaluation are optimized for processing 10 seconds of single-lead ECG signals, each strip has been labeled regarding the presence/absence of any abnormal beat within the time strip. We have repeated 10 times a 10-fold cross-validation across the 78 recordings, where a SVC is fitted on top of the representations. CuPID performs significantly better compared to the baselines as shown in Table 1.

Table 1: Performance metrics for the different downstream tasks. Bold and underline values represent the best and the second-best performances, respectively.

| | MIT-AFIB | | LT-AF | | SVT | |
|---|---|---|---|---|---|---|
| | Accuracy | F1 | Accuracy | AUROC | Accuracy | AUROC |
| PCLR | 0.752 | 0.738 | $0.808 \pm 0.003$ | $0.801 \pm 0.006$ | $0.493 \pm 0.014$ | $0.586 \pm 0.010$ |
| CLOCS | 0.664 | 0.590 | $0.678 \pm 0.010$ | $0.766 \pm 0.014$ | $0.520 \pm 0.008$ | $0.561 \pm 0.011$ |
| DEAPS | 0.763 | 0.747 | $0.843 \pm 0.005$ | $0.882 \pm 0.007$ | $0.483 \pm 0.014$ | $0.578 \pm 0.010$ |
| Mix-Up | 0.619 | 0.569 | $0.610 \pm 0.008$ | $0.648 \pm 0.017$ | $0.526 \pm 0.011$ | $0.612 \pm 0.010$ |
| MTAE | 0.766 | 0.73 | $0.805 \pm 0.006$ | $0.884 \pm 0.006$ | $0.512 \pm 0.006$ | $0.603 \pm 0.009$ |
| Jepa | 0.751 | 0.705 | $0.781 \pm 0.005$ | $0.868 \pm 0.004$ | $0.523 \pm 0.006$ | $0.621 \pm 0.007$ |
| CuPID | **0.863** | **0.843** | **$0.879 \pm 0.003$** | **$0.934 \pm 0.002$** | **$0.580 \pm 0.0122$** | **$0.660 \pm 0.005$** |

It is important to note that MTAE is essentially CuPID without the spectrogram, thus this evaluation indirectly demonstrates the enhancement brought by incorporating the spectrogram. However, this effect will be examined in greater detail in the ablation studies (see Section 5).

## 4.2 BENCHMARKING CuPID IN PTB-XL AND CPSC-2018

The aim of CuPID is to generate meaningful representations of single-lead ECG data. Consequently, the primary experiment was conducted on databases where the signals were recorded by a Holter monitor. Nonetheless, we have evaluated CuPID on widely-used benchmarked datasets such as *PTB-XL* (Wagner et al., 2020), and *CPSC2018* (Alday et al., 2021), that consist of 10 seconds 12-lead ECG signals recorded in clinical setup. The methods that compel the set of baselines for these two experiments are the following; MoCO v3 (Chen et al., 2021); Contrastive Multi-segment Coding (CMSC) from (Kiyasseh et al., 2021); MTAE, Masked Lead AutoEncoder (MLAE) from (Zhang et al., 2023); and ST-MEM (Na et al., 2024) The architecture employed by these methods consists of an encoder with 12 blocks with 768 dimensions trained on 12-Lead ECG data.

While all the baselines included in this experiment utilize the available 12 leads, CuPID only processes the II lead, being the one closer to the signal recorded by the Holter monitor. We want to highlight that, as shown in Table 3, CuPID achieves the second-best metrics on these two benchmarked datasets. We consider this achievement of significant relevance considering CuPID only uses one ECG lead sampled with a lower resolution, a significantly smaller model trained (4 blocks and 128 dimensions) on a noisy database.

Table 2: Performance Metrics PTB-XL and CPSC2018. * means that scores are given based on the ST-MEM (Na et al., 2024) work. Bold and underline values represent the best and the second-best performance, respectively.

| | PTB-XL | | | CPSC2018 | | |
|---|---|---|---|---|---|---|
| | Accuracy | F1 | AUROC | Accuracy | F1 | AUROC |
| MoCo v3* | $0.552 \pm 0.000$ | $0.142 \pm 0.000$ | $0.739 \pm 0.006$ | $0.268 \pm 0.055$ | $0.080 \pm 0.038$ | $0.712 \pm 0.054$ |
| CMSC* | $0.681 \pm 0.032$ | $0.441 \pm 0.058$ | $0.797 \pm 0.038$ | $0.361 \pm 0.005$ | $0.238 \pm 0.022$ | $0.724 \pm 0.013$ |
| MTAE* | $0.683 \pm 0.008$ | $0.437 \pm 0.012$ | $\underline{0.807 \pm 0.006}$ | $0.486 \pm 0.012$ | $0.349 \pm 0.034$ | $0.818 \pm 0.010$ |
| MTAE + RLM* | $0.687 \pm 0.006$ | $0.444 \pm 0.009$ | $0.806 \pm 0.005$ | $0.480 \pm 0.010$ | $0.342 \pm 0.022$ | $0.824 \pm 0.006$ |
| MLAE* | $0.649 \pm 0.008$ | $0.382 \pm 0.020$ | $0.779 \pm 0.008$ | $0.443 \pm 0.014$ | $0.263 \pm 0.021$ | $0.794 \pm 0.016$ |
| ST-MEM* | $\mathbf{0.726 \pm 0.005}$ | $\mathbf{0.508 \pm 0.008}$ | $\mathbf{0.838 \pm 0.011}$ | $\mathbf{0.723 \pm 0.008}$ | $\underline{0.641 \pm 0.010}$ | $\mathbf{0.938 \pm 0.002}$ |
| CuPID | $\underline{0.710 \pm 0.011}$ | $\underline{0.487 \pm 0.011}$ | $0.800 \pm 0.010$ | $\underline{0.685 \pm 0.001}$ | $\mathbf{0.650 \pm 0.001}$ | $\underline{0.928 \pm 0.000}$ |

We would like to highlight that CuPID, through the integration of the spectrogram, significantly enhances performance compared to its counterpart, MTAE, even though it processes 12 leads while CuPID processes only one lead. For the sake of clarity, we have not included the single-lead ECG baselines, since the better performance of CuPID has been assessed on the previous experiments. However, we have provided the corresponding table in the Appendix (See Section A).

### 4.2.1 DISCUSSION OF THE RESULTS

Throughout this comprehensive evaluation, it has been established that by cueing the predictor, CuPID drives the learning encoder to compute more detailed patch representations. This results in the model achieving markedly enhanced results in a variety of downstream tasks, as detailed in Table 1. These findings provide robust evidence in favor of the hypotheses posited by this study: (i) The predictor's inability to reconstruct the original signal due to the unpredictability of the delay limits the learning potential of the encoder. (ii) By cueing the predictor with the spectrogram, we enable it to deal with this delay and drive the encoder to compute detailed token representations that can be used to reconstruct the original input with a great level of detail. (iii) The more informative the patch representations are, the more informative the class token will be, improving the performance of the model when addressing downstream tasks.

## 5 ABLATION AND SENSITIVITY STUDIES

To assess the primary technical innovation of CuPID, specifically the integration of the spectrogram into the predictor, a comprehensive ablation study was conducted. This study examined the model's performance improvements across the two most widely used benchmarks (*PTB-XL* and *CPSC2018*), considering various masking ratios during pre-training. Figure 5 not only justifies the the choice of 0.5 as the value for the random masking hyperparameter, more importantly, it proves that the increase in performance when adding the spectrogram is consistent across tasks and masking ratios.

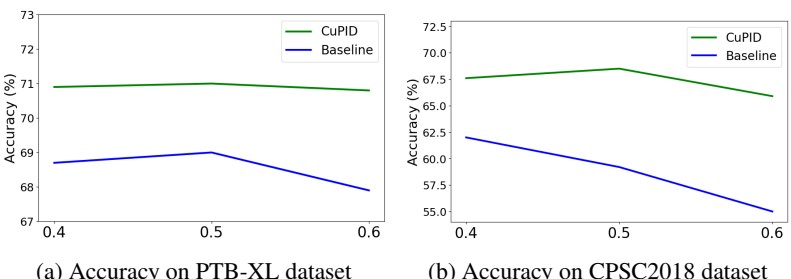

(a) Accuracy on PTB-XL dataset      (b) Accuracy on CPSC2018 dataset

Figure 5: Effect of incorporating the spectrogram for input reconstruction and different mask ratios.

This paper has validated CuPID's decision to reconstruct the original input instead of the teacher network's representations. Additionally, the evaluation results (refer to Section 4) endorse this choice. Nonetheless, the impact of integrating the spectrogram has also been analyzed within this alternative framework, considering various masking ratios. Figure 6 demonstrates that adding the spectrogram to the predictor also carries out benefits to this alternative framework.

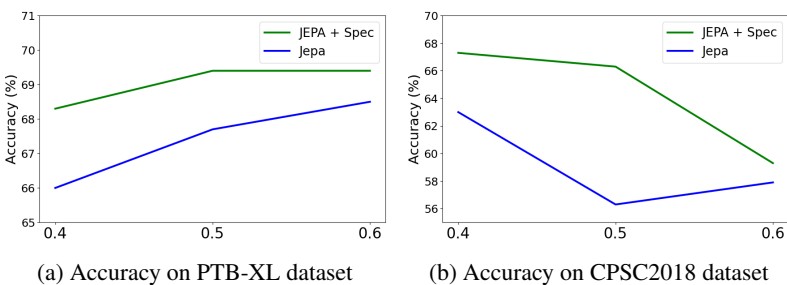

(a) Accuracy on PTB-XL dataset      (b) Accuracy on CPSC2018 dataset

Figure 6: Effect of incorporating the spectrogram for I-JEPA approach and different mask ratios.

The impact of incorporating the spectrogram during pre-training on a different database, specifically Icentia 11K, has been assessed. Due to the higher noise levels in Icentia 11K compared to SHHS, the results are less favorable, as illustrated in Figure 7. This figure not only supports the selection of SHHS as the primary database for the evaluation but also demonstrates the advantages of using the spectrogram in noisier environments.

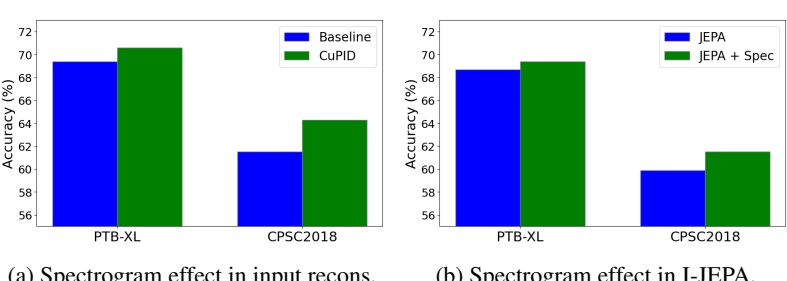

(a) Spectrogram effect in input recons.      (b) Spectrogram effect in I-JEPA.

Figure 7: Ablation study in Icentia dataset.

## 6 CONCLUSION

This research provides strong evidence that directly applying the Masked Data Modelling (MDM) framework to single-lead ECG signals is insufficient. This is due to the idiosyncrasies of ECG data, where consecutive data chunks represent a distinct wave, and the distance between consecutive heartbeats varies moderately. This leads the predictor to be cautious when reconstructing the masked patches and to not drive the encoder to compute detailed patch representations that can be used for addressing downstream tasks. To overcome this issue, we introduce CuPID, a novel SSL technique for ECG analysis. By cueing the predictor with the contextual information given by the spectrogram of the input signal, CuPID enforces the encoder to compute more informative representations. It results in a significant performance improvement when addressing downstream tasks.

**Limitations:** CuPID has only been evaluated on a single architecture configuration. However, the incorporation of the spectrogram in the predictor is agnostic to the ViT configuration and similar performance improvements should be obtained .

**Future Work:** We were surprised to observe a decline in performance when pre-training the model on the Icentia 11K database, despite it being theoretically more comprehensive than SHHS. We believe this issue stems from the high level of noise present in the Icentia 11K database. Moving forward, we aim to explore potential integrations with CuPID to address this problem and fully leverage the database's potential.

## 7 REPRODUCIBILITY STATEMENT

The attached code as a part of the supplementary material encompasses the implementation of CuPID and several other baselines. Moreover, comprehensive details on training hyperparameters, schemes, and hardware specifications are provided. In addition the pseudocode for the method is provided in the Appendix. Finally, we furnish the pre-trained model's parameters to facilitate others in achieving reproducible results, together with the code used for processing each database.

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

## A EVALUATION OF SINGLE-LEAD ECG BASELINES IN PTB-XL AND CPSC2018

Table 3: Performance Metrics PTB-XL and CPSC2018

|  | PTB-XL | | | CPSC2018 | | |
|---|---|---|---|---|---|---|
|  | Accuracy | F1 | AUROC | Accuracy | F1 | AUROC |
| CLOCS | $0.647 \pm 0.012$ | $0.385 \pm 0.016$ | $0.755 \pm 0.011$ | $0.471 \pm 0.002$ | $0.414 \pm 0.001$ | $0.827 \pm 0.000$ |
| PCLR | $0.679 \pm 0.010$ | $0.446 \pm 0.015$ | $0.777 \pm 0.012$ | $0.631 \pm 0.002$ | $0.594 \pm 0.003$ | $0.903 \pm 0.000$ |
| DEAPS | $0.7000 \pm 0.011$ | $0.476 \pm 0.000$ | $0.796 \pm 0.001$ | $0.667 \pm 0.002$ | $0.634 \pm 0.002$ | $0.918 \pm 0.002$ |
| Mix-Up | $0.660 \pm 0.011$ | $0.420 \pm 0.017$ | $0.760 \pm 0.012$ | $0.502 \pm 0.002$ | $0.451 \pm 0.004$ | $0.837 \pm 0.000$ |
| MTAE | $0.690 \pm 0.011$ | $0.462 \pm 0.16$ | $0.794 \pm 0.012$ | $0.593 \pm 0.002$ | $0.543 \pm 0.002$ | $0.894 \pm 0.000$ |
| Jepa | $0.677 \pm 0.010$ | $0.445 \pm 0.017$ | $0.774 \pm 0.010$ | $0.563 \pm 0.001$ | $0.514 \pm 0.002$ | $0.880 \pm 0.000$ |
| CuPID | $0.710 \pm 0.011$ | $0.487 \pm 0.015$ | $0.800 \pm 0.010$ | $0.685 \pm 0.001$ | $0.650 \pm 0.001$ | $0.928 \pm 0.000$ |

## B DETAILS OF DATASETS USED FOR MAIN EVALUATION OF SINGLE-LEAD ECG BASELINES

Table 4: MIT-BIH Atrial Fibrillation (MIT-AFIB)

| Label | # ECGs | # Record. Count & (Ratio) | Ratio #ECGs per Record. |
|---|---|---|---|
| Normal Sinus Rhythm (NSR) | 50115 | 21 (91.3%) | $0.401 \pm 0.357$ |
| Atrial Fibrillation (AFib) | 33694 | 23 (100%) | $0.656 \pm 0.320$ |

Table 5: Long Term AF (LT-AF)

| Label | # ECGs | # Record. Count & (Ratio) | Ratio #ECGs per Record. |
|---|---|---|---|
| Normal Sinus Rhythm (NSR) | 270702 | 53 (63.1%) | $0.672 \pm 0.315$ |
| Atrial Fibrillation (AFib) | 368272 | 84 (100%) | $0.546 \pm 0.422$ |
| Bradycardia | 19197 | 35 (41.7) | $0.072 \pm 0.100$ |

Table 6: MIT-BIH Supraventricular Arrhythmia (MIT-SVA)

| Label | # ECGs | # Record. Count & (Ratio) | Ratio #ECGs per Record. |
|---|---|---|---|
| Normal Sinus Rhythm (NSR) | 6608 | 76 (97.4%) | $0.296 \pm 0.300$ |
| Ventricular Beats | 2184 | 70 (89.7%) | $0539 \pm 0.316$ |
| Supraventricular Beats | 2543 | 62 (79.5%) | $0.267 \pm 0.287$ |

## C DATA PREPROCESSING

To ensure the complete reproducibility of this work, this section presents a detailed description of the preprocessing steps employed for the training and evaluation databases utilized in the proposed method.

## C.1 SHHS DATA SELECTION

Only the subjects that appear in both recording cycles are used during the training procedure. This leads to 2643 subjects. ECG signals are extracted from the Polysomnography (PSG) recordings. The quality of every 10 seconds-data strips has been evaluated with the algorithm proposed by Zhao and Zhang Zhao & Zhang (2018). We use SHHS since it contains two records belonging to the same subject. This makes this specific database special, and this is the reason that it has been the only database used during the optimization.

## C.2 DATA CLEANING

In addition, all signals from the utilized datasets were resampled to a frequency of 100Hz. Then, a $5^{th}$ order Butterworth high-pass filter with a cutoff frequency of 0.5Hz was applied to eliminate any DC-offset and baseline wander. Finally, each dataset underwent normalization to achieve unit variance, ensuring that the signal samples belong to a $\mathcal{N}(0,1)$ distribution. This normalization process aimed to mitigate variations in device amplifications.

## C.3 PSEUDOCODE

---

**Algorithm 1:** Cueing the Predictor Increments the Detailing (CuPID)

---

**Input:**

    $K$ and $B$             $\triangleright$ Number of iterations and Batch Size

    $\mathcal{F}(x)$ and $\mathcal{P}(h, s)$            $\triangleright$ Encoder and Predictor

    $\theta$ and             $\triangleright$ Trainer Parameters and Optimizer

    $\mathcal{S}(x)$             $\triangleright$ Spectrogram Transorm

    $\mathcal{RM}(X)$             $\triangleright$ Random Mask Function

    $\mathcal{R}ec(h, \mathcal{M}_t)$            $\triangleright$ Attach Mask tokens for Predictor Input

    $\mathcal{M}_t$             $\triangleright$ Learnable Mask Token

    $\mathcal{L}_1(X, Y, M)$            $\triangleright$ L1 Loss Functions

**for** $k \leftarrow 0$ **to** $K$ **do**

    $X \leftarrow \{X^1 \cdots X^N\}_{b=0}^B$       $\triangleright$ Sample $N$ inputs from dataset

    $H_m, M \leftarrow \mathcal{RM}(X)$       $\triangleright$ Random Masking and get Mask Matrix

    $H_m \leftarrow \mathcal{F}_{(h_m)}$       $\triangleright$ Encoder Representations

    $H \leftarrow \mathcal{R}ec(h_m, \mathcal{M}_t)$       $\triangleright$ Attach mask tokens for Predictor's input

    $S \leftarrow \mathcal{S}(X)$       $\triangleright$ Compute the Spectrogram

    $Y \leftarrow \mathcal{P}(h, s)$       $\triangleright$ Compute Predictor Reconstructions

    $l \leftarrow \mathcal{L}_1(X, Y, M)$       $\triangleright$ L1 Loss on masked patches

    $\partial\theta \leftarrow \partial_\theta l$       $\triangleright$ Compute loss gradients for $\theta$

    $\theta \leftarrow opt(\theta, \partial_\theta)$       $\triangleright$ Update the Parameters

**end**

---

---

**Algorithm 2:** CuPID's Predictor

---

**Input:**

    $\mathcal{P}, \mathcal{O}(H)$       $\triangleright$ Predictor and Final Layer

    H, S       $\triangleright$ Predictor Input and Spectrogram

**for** $idx, \mathcal{P}_l$ $in$ $enum(\mathcal{P})$ **do**

    **if** $idx = 0$ **then**

        $H \leftarrow \mathcal{P}_l(H, H, H)$;

    **else**

        $H \leftarrow \mathcal{P}_l(H, S, H)$       $\triangleright$ Fed the Sectrogram as the Key

    **end**

**end**

$Y \leftarrow \mathcal{O}(H)$ return $Y$;

---

