# OpenReview forum: "CuPID: Leveraging Masked Single-Lead ECG Modelling for Enhancing the Representations"
_ICLR.cc/2025/Conference — ICLR 2025 Conference Withdrawn Submission_

### Official Review · Reviewer_Frou · 2024-10-25

**Soundness:** 2
**Presentation:** 2
**Contribution:** 2
**Rating:** 3
**Confidence:** 5

**Summary:**

This paper provides a pre-training framework (CuPID) for single-lead ECG signals that leverages masked data modeling (MDM), with the proposed novel idea of using ECG signal’s spectrogram as the key for the decoder Transformer. They pre-train CuPID on “SHHS” dataset containing ECG signals from 2463 subjects, and evaluate the trained model on various single-lead and 12-lead ECG datasets such as “MIT-AFIB”,  “LT-AF”, “MIT-SVA”, “PTB-XL” and “CPSC2018”. They provide results against several baselines and perform some ablations as well.

**Strengths:**

* The idea of using spectrograms only as part of the network (part of input to the decoder Transformer) is novel and interesting.
* I appreciate that the authors used several different datasets to evaluate their model.
* I also appreciate that the authors compared their method with several baselines.

**Weaknesses:**

I believe authors can be improve the paper on various fronts including the rigor of the analyses/claims, the level of details in the paper and presentation.

* Authors have not acknowledged/discussed very important related body of literature related to modeling single-lead ECG and masked data modeling (MDM) [1-4]. In addition, claims such as “However, it is impractical to directly apply these [MDM] methods to single-lead ECG data.” are not clear and backed by the literature to me. Particularly, masked autoencoders (as introduced in He et al. 2021, and used in these follow up works) can be applied to arbitrary number of channels in input, whether single-lead or 12-lead ECG. It would be great if the authors discuss the relevant work, and provide more emprical/theoretical evidence on why “it is impractical to directly apply these [MDM] methods to single-lead ECG data.”.
    * [1], [2], [3] just a few examples of using MDM for ECG
    * [4] for using single-lead ECG.


* Significant details about how the baseline models/methods were trained are missing from the paper. Authors mention that “To ensure fairness in the evaluation, all the methods have been trained using the same training configuration, encoder, and dataset as CuPID.” but the baseline methods have significantly different set of hyperparameters, and it’s not clear how authors picked those, and how they did tuning to ensure fair comparisons. For example, JEPA has different hyperparameters like the selection of the context/target windows/blocks, just to name one, or contrastive methods, like CLOCS and PCLR, have different hyperparameters, such as temperature in the contrastive loss, just to name one. I believe authors can do a more rigorous job optimizing baselines and explaining the details, to demonstrate fairness in comparisons.
* The presentation throughout the manuscript can be significantly improved, which would help the manuscript’s quality. Here, I provide few examples, and authors may consider fixing or adding explanations:
    * Overall, figures can aesthetically improve. Just to name 2 examples:
        *  Authors have used various font sizes (and probably even fonts) in Figure 4.
        * It is not clear to me whether in Figure 4, their decoder Transformer does not have skip connections (if yes, why?) or they’re missing from the figure. Please consider explaining or adding.
    * Figure 1 is just adopted from other works and contains well-known information about ECG; I am not sure if it’s worth re-iterating such obvious points as a Figure 1 of the paper (perhaps appendix is a better place to put such information).
    * Inconsistent name for JEPA in Figure 6.
    * Panels b and c from Figure 2 can be combined to provide users for a better visual representation of improvements
    * Figures 2 and 3 can be combined.
* The details of evaluations are missing from the paper:
    * The choice of cross-validation across datasets seems arbitrary. For example, for “MIT-AFIB“ authors have done leave-one-subject-out, but for ”LT-AF“ they have done 10-fold cross-validation. It would be great if they can explain why these selections are different.
    * The choice of predictor models seems arbitrary. For example,  for “MIT-AFIB“ authors have used SVM, but for ”LT-AF“ they have used logistic regression. It would be great if they can explain why these selections are different.
    * It is unclear why in “MIT-AFIB” and “LT-AF”, the cross-validation is done across subjects, but in “MIT-SVA”, they do the cross-validation across recordings. If each participant has 1 recording in “MIT-SVA”, it makes sense, in that case please clarify.
    * See question 5 below as well.
* L151-L153: These claims are not supported. It appears to me that the authors make several hypotheses/claims ruling out pre-training frameworks that do not reconstruct the signal in the output space, although they are popularly used in other domains and even for health applications. Claims such as “This interesting property does not occur when reconstructing the representations from the teacher network, since they are expected to lie with the same range of values.” requires theoretical or comprehensive empirical backup.
* Several typos throughout the manuscript can be fixed, just to name a couple examples:
    * 1) L149:  “we consider a more suitable to”
    * 2) L80: missing space
    * 3) L367: header in Table 1 does not match the acronym in L357.

- Several other points are not clear that I asked in the questions below.

[1] Sawano, Shinnosuke, et al. "Applying masked autoencoder-based self-supervised learning for high-capability vision transformers of electrocardiographies." Plos one 19.8 (2024): e0307978.

[2] Song, Junho, et al. "Foundation Models for Electrocardiograms." arXiv preprint arXiv:2407.07110 (2024).

[3] McKeen, Kaden, et al. "Ecg-fm: An open electrocardiogram foundation model." arXiv preprint arXiv:2408.05178 (2024).

[4] Attia, Zachi I., et al. "Prospective evaluation of smartwatch-enabled detection of left ventricular dysfunction." Nature medicine 28.12 (2022): 2497-2503.

**Questions:**

*  I have a question about normalization statement “Finally, each dataset underwent normalization to achieve unit variance, ensuring that the signal samples belong to a N(0,1) distribution. This normalization process aimed to mitigate variations in device amplifications.”; is this at sample-level or at dataset-level, and can authors provide more information on how it’s done?
* Figure 5/6: can the authors please provide the results for wider masking ratios? It is typical in masked auto encoding that larger ratios are masked (see He et al, 2021).
* Why did the authors choose L1 loss over L2 loss, which is more standard in masked auto encoders (He et al, 2021)? Similar question for their baselines.
* Can authors please provide number of subjects in Appendix Section B?
* From loss curves in Figure 2, it appears that there’s still room for improvements in both training frameworks. What mechanism did the authors use to stop the training, and why training was stopped at 40K iterations?
* What does “#ECGs”, “# Record. Count”, and “Ratio” mean in Table 4-6?
* Curious why authors have used `Rec` for “Attach mask tokens” in “Algorithm 1”? Also, to be frank, I found Algorithms 1 and 2 unnecessarily complicated and without additional useful information. Authors may consider improvements such as using a pytorch-style code (not necessary, just a suggestion), or any other way of their preference.

---

### Official Review · Reviewer_QXvT · 2024-10-28

**Soundness:** 2
**Presentation:** 3
**Contribution:** 3
**Rating:** 5
**Confidence:** 3

**Summary:**

The paper proposes to feed spectrogram information for masked signal modeling, and the experiments show better performance on some downstream prediction tasks.

**Strengths:**

1. The paper is well written, and easy to read.
2. The proposed method is interesting, and shows better performance based on limited evaluations settings.

**Weaknesses:**

1. Figure 3 seems the main figure of the paper, and I’ll suggesting moving it to the first/second page. Page 5 is too late.
2. Formula 1 seems un-necessary and does not add much value.
3. \[Line 262\] Why spectrogram information was introduced in the second block is not well justified.
4. Formula 2 seems unnecessary as well, as people should generally know what L1 loss is, and it sounds common sense to not backpropagate on unmasked tokens.
5. \[Line 310-312\] There’re many decisions made here without any justification. For example, the episode length, sampling frequency and patching are also important decisions which might affect performance significantly.
6. On top of 5, (as acknowledged in the limitation), this study only considers one configuration, which is unreasonable given the relatively small size of the model.
7. The benchmark datasets are limited in terms of downstream tasks.
8. The ablation study is highly limited. For example, can we introduce the spectrogram through another method (via Q rather than K, or concatenation instead of elementwise addition)? How does it affect the performance?

**Questions:**

1. \[Line 053\] Can you explain how Figure 2b shows the model prefers predicting the average rather than the detailed waveform?
2. Further on this, does more detailed fitting of waveform morphology lead to better clinical interpretation? Is there any trade-off here?
3. What’s the X-axis of Figure 2a? \# of training steps? What’s “iteration” here?
4. \[Line 259\] Can you further explain “ensuring the spectrogram remains unmasked”?

---

### Official Review · Reviewer_cZCp · 2024-11-04

**Soundness:** 1
**Presentation:** 1
**Contribution:** 2
**Rating:** 1
**Confidence:** 5

**Summary:**

The paper introduces "CuPID" (Cueing the Predictor Increments the Detailing), a novel self-supervised learning (SSL) method designed to enhance single-lead ECG data analysis. Recognizing the unique challenges of single-lead ECG signals, CuPID adapts the Masked Data Modelling (MDM) framework by cueing the predictor with spectrogram-derived context, which enables the model to generate richer, more detailed representations. This approach allows the predictor to better handle the time-variant intervals in ECG signals, overcoming limitations of traditional MDM methods.

Key contributions include:
A SSL method for single-lead ECGs that leverages spectrogram-based cues for improved data representation.

**Strengths:**

The development of robust methodologies for analyzing single-lead ECG is necessary in this field.

**Weaknesses:**

Since there has been significant prior research on generating spectrograms from ECG and utilizing them alongside raw waveforms, the approach of adding spectrograms of patch-based waveforms as additional inputs in the MDM approach is not particularly novel (https://arxiv.org/pdf/1812.05555, https://www.mdpi.com/2076-3417/14/21/9936).

- Lack of Soundness
The paper lacks a clear explanation of how tasks were evaluated. For instance, in the case of MIT-AFIB, there are segments with and without AFib. How were Accuracy and F1 calculated in this context? What criteria were used to determine True Positives?

The ablation study does not seem to consider a sufficiently wide range of parameters. Furthermore, the benchmarks used for comparison in the ablation study appear to be too weak. Given the performance of other methods in Table 2, the benchmarks used in the ablation study do not seem appropriate.
Descriptions of the figures are insufficient, with missing information about the ECG waveform and the lack of error bars in the ablation study results.

- Insufficient Performance
The model’s accuracy and F1 score seem insufficient, especially considering that other models for diagnosing AFib on the MIT-BIH Arrhythmia dataset have achieved very high levels of performance.

In the benchmarking on PTB-XL and CPSC2018, CuPID showed lower performance compared to ST-MEM. Given that ST-MEM is a method applicable to both single-lead and multi-lead ECG, this raises questions about the utility of CuPID.

**Questions:**

Improvements are needed in the areas outlined in the weaknesses section above.

---

### Official Review · Reviewer_ccTU · 2024-11-04

**Soundness:** 2
**Presentation:** 1
**Contribution:** 1
**Rating:** 1
**Confidence:** 5

**Summary:**

The CuPID predict receives additional information(spectogram cues) to reconstruct the original signal, which returns more informative class tokens.

**Strengths:**

By using the ECG specific information such as R peaks and the spectogram of signals, this paper seeks to reconstruct the original ECG more than other baseline methods.

**Weaknesses:**

1. **Insufficient Explanation of the Clinical Challenge of ECGs and Motivation of its approach**:
   - The paper lacks a clear explanation of the clinical challenge it addresses, particularly what is meant by "idiosyncrasies." Additionally, the phrase "this clinical challenge" in the last sentence of the first paragraph of the introduction is ambiguous and does not clarify the specific challenge being referenced.
   - It is unclear why the proposed method is specifically tailored for single lead data.

2. **Lack of Experiments to Validate R-R interval Accuracy**:
   - There are insufficient experiments to confirm whether the proposed method accurately follows the R-R interval.

3. **Figure Similarity to Existing Literature**:
   - Figure 4 closely resembles the diagram in the "Attention is All You Need(Vaswani et al., 2017)" paper, suggesting that it may need to be redrawn to differentiate it from existing works.

4. **Lack of Novelty**:
   - The proposed method appears to be merely an extension of existing MDM techniques by incorporating attention on spectrograms, which raises concerns about its novelty.

**Questions:**

1. **Lack of Connection Between Motivation and Methods**:
   - There is insufficient explanation regarding how the use of a spectrogram is beneficial for finding R-R intervals. Could you elaborate on this connection?

2. **Inconsistency in Experimental Results Tables**:
   - In Table 1, why do some metrics report F1 scores while others report AUROC? Can you clarify the rationale behind this inconsistency?

3. **Confusion of the Use of Terminology**:
   - Does the R-R interval refer to the temporal distance between adjacent R peaks?
   - In Figure 1.(b), the ECG appears to have a consistent R-R interval. Moreover, the ECG shown in Figure 1.(b) is labeled as "Normal ECG," and one clinical characteristic of a Normal ECG is a consistent R-R interval.
   - Please also provide the y-scale in Figure 1.(c).

---

### Note · Authors · 2025-01-29

I have read and agree with the venue's withdrawal policy on behalf of myself and my co-authors.